# Improving Our Understanding and Practice of Antibiotic Prescribing: A Study on the Use of Social Norms Feedback Letters in Primary Care

**DOI:** 10.3390/ijerph18052602

**Published:** 2021-03-05

**Authors:** Stephanie Steels, Natalie Gold, Victoria Palin, Tim Chadborn, Tjeerd Pieter van Staa

**Affiliations:** 1Department of Social Care and Social Work, Faculty of Health, Psychology and Social Care, Manchester Metropolitan University, Manchester M15 6GX, UK; 2Public Health England Behavioural Insights, Wellington House, 133-155 Waterloo Road, London SE1 8UG, UK; Natalie.Gold@phe.gov.uk (N.G.); Tim.Chadborn@phe.gov.uk (T.C.); 3Faculty of Philosophy, University of Oxford, Radcliffe Humanities, Woodstock Road, Oxford OX2 6GG, UK; 4Centre for Health Informatics, Division of Informatics, Imaging and Data Science, School of Health Sciences, Faculty of Biology, Medicine and Health, The University of Manchester, Manchester M13 9YP, UK; victoria.palin@manchester.ac.uk (V.P.); tjeerd.vanstaa@manchester.ac.uk (T.P.v.S.); 5Utrecht Institute for Pharmaceutical Sciences, Utrecht University, Heidelberglaan 8, 3584 CS Utrecht, The Netherlands

**Keywords:** antibiotics, social norms, prescribing behaviour, feedback, primary care

## Abstract

In the UK, 81% of all antibiotics are prescribed in primary care. Previous research has shown that a letter from the Chief Medical Officer (CMO) giving social norms feedback to General Practitioners (GPs) whose practices are high prescribers of antibiotics can decrease antibiotic prescribing. The aim of this study was to understand the best way for engaging with GPs to deliver feedback on prescribing behaviour that could be replicated at scale; and explore GP information requirements that would be needed to support prescribing behaviour change. Two workshops were devised utilising a participatory approach. Discussion points were noted and agreed with each group of participants. Minutes of the workshops and observation notes were taken. Data were analysed thematically. Four key themes emerged through the data analysis: (1) Our day-to-day reality, (2) GPs are competitive, (3) Face-to-face support, and (4) Empowerment and engagement. Our findings suggest there is potential for using behavioural science in the form of social norms as part of a range of engagement strategies in reducing antibiotic prescribing within primary care. This should include tailored and localised data with peer-to-peer comparisons.

## 1. Introduction

The extensive use of antibiotics in both human medicine and animal agriculture, combined with the slow development of new compounds to treat infections has led to a rapid evolution of Antimicrobial Resistance (AMR) [1]. If no action is taken to preserve the effectiveness of existing antibiotics, resistance evolution will lead to the increase of the burden on health services and increase infection-related mortality worldwide [2]. In the U.K, 81% of all antibiotics are prescribed in primary care [3]. However, modelling has shown that up to 23.1% of antibiotic prescribing is inappropriate, either having no benefit to the patient or having a marginal benefit that is outweighed by potential risks [4].

Behavioural sciences are increasingly being used to improve the effectiveness of interventions by focusing on understanding and changing behaviours [5,6]. Social norms are a way of influencing people’s behaviour by utilising indirect suggestions or interventions as ways to (positively) influence the behaviour and decision making of individuals or a group of people (such as the general population) by highlighting the compliance of the majority [7].

Since 2014, an annual feedback letter from the Chief Medical Officer (CMO) has been sent to general practices, giving feedback on their prescribing compared to other practices [8,9]. The content of the letter was based on a comprehensive review and analysis of the behaviours that support antibiotic stewardship, and which of the drivers of these behaviours are amenable to change [8].

All letters contain social norms feedback. In the first year, this was a comparison to other prescribers in local area teams: ‘The great majority (80%) of practices in [local NHS area team] prescribe fewer antibiotics per head than yours’ (see Figure 1 for a picture of the letter). However, local area teams were disbanded and in subsequent years GPs were given feedback compared to the national data [10]: ‘The great majority (80%) of practices in England prescribe fewer antibiotics per head than yours’. GPs were directed to their prescribing data on Fingertips, https://fingertips.phe.org.uk/profile/amr-local-indicators (accessed on 29 June 2020). The antimicrobial prescribing indicators are updated quarterly, appearing roughly three months after the end of the quarter they cover. This is due to the time lag in receiving the data from NHS Business Authority and the time it takes to quality check, process and upload the data [11]. Furthermore, the NHS Business Authority currently only collects prescribing data at the level of the GP practice, and not the individual GP, so feedback has always been of practice-level prescribing.

All letters contain simple steps to reduce prescribing, and a copy of the ‘TARGET treat your infection’ leaflet was enclosed with the letter as an aid for patient conversations. Letters were usually sent to land at the beginning of September, however, for pragmatic reasons that has not always been possible, but they have always been sent during the UK flu season (September–March) since that is the time of year with the highest antibiotic prescribing. In 2018/9 and 2019/20, as a part of Randomized Controlled Trials to see if the content of the letter could be improved, some practices received letters with charts, showing their previous year’s prescribing compared to the average prescribing of their peers, where the annual prescribing data were based on the most recent 12-months of prescribing available at the time of producing the letter [3].

This study investigated the effectiveness of current feedback letters and explored what types of GP level prescribing data could be used in future feedback interventions. The aim of this research was twofold: (1) to understand the best way for engaging with GPs in discussing prescribing behaviour that could be replicated at scale; and (2) to explore the types of GP prescribing data that would be used to support changes to prescribing behaviour change.

## 2. Methods

This study was based on qualitative data gathered through two workshops held with GPs, other prescribers and practice managers who work in primary care. Data were collected between February and March 2020.

### 2.1. Data Collection

The research team consisted of an independent researcher from Manchester Metropolitan University (S.S.) and the project team (F.J., V.P., G.T., T.P.v.S. and E.T.) of the project Building Rapid Interventions to reduce antibiotic resisTance (BRIT) and Public Health England Behavioural Insights (PHEBI), who commissioned the research (N.G. and T.C.). The BRIT project collects and analyses electronic healthcare data from participating primary care practices, interprets the results and then presents these insights to the health professionals [https://www.britanalytics.uk/] (accessed on 15 February 2020).

A participatory approach, whereby a group of people are brought together to seek their opinions, extract their knowledge and to solve problems in a collaborative and creative environment, was used to inform the design of the workshop [12]. The workshop covered the following areas: three feedback letters developed by PHEBI that were presented during the workshop (see Figure 2, Figure 3 and Figure 4), how GPs would want to receive information on their own prescribing practice, presentation of prescribing data and mechanisms for facilitating behaviour change. A topic guide was developed to explore and capture information on these areas. Each workshop lasted three hours in an afternoon, with one workshop held in Manchester and one in Nottingham. A third workshop was cancelled due to COVID-19 lockdown measures.

### 2.2. Sampling and Recruitment

The specific aim of the study resulted in employing a purposive sampling strategy [13]. Participants were eligible for inclusion if they were a General Practitioner (GP) or other type of prescriber (such as pharmacists) based in either Nottingham or Manchester. Participants were recruited to the workshops by the BRIT team via email from existing BRIT dashboard users. A follow-up email was sent two weeks later. Non-BRIT dashboard users were recruited through the Clinical Research Network (CRN). In total, 85 GP practices were invited to take part in the study. As recruitment progressed, the inclusion criteria were widened to include practice managers from both BRIT dashboard and non-BRIT dashboard using practices to ensure a minimum of ten and a maximum of twenty participants were able to attend each workshop. In total, fifteen participants were recruited for each workshop.

It was important that the workshops created an environment to enable all participants an opportunity to share their experiences and opinions on each of the areas that would be explored. In addition, it was important that the workshop provided a ‘safe space’ where participants would feel at ease to be open and free with their opinions [14]. Participants were encouraged to write down thoughts, comments and suggestions on post-it notes allocated on each table. Discussions were facilitated by the independent researcher (S.S.) with support from the BRIT team (F.J., V.P., G.T., T.P.v.S. and E.T.) in exploring prescribing data presentation. Key discussion points were noted and agreed on flip chart paper with each group of participants [13]. Furthermore, minutes of the workshops and observation notes were taken in each workshop. Participants were asked to email any further comments to the independent researcher on leaving the workshop. Earlier tests of the recording equipment found that it was not possible to make a recording of suitable quality for transcription therefore no audio recordings were made during the workshops due to the size of the room and the number of participants.

### 2.3. Data Analysis

Data gathered by S.S., V.P., F.J. and G.T. were combined into one Word document, with S.S. carrying out additional checks to ensure accuracy in data entry. The data were then exported into NVivo to be thematically analysed. Thematic analysis is a widely used method in analysing qualitative research data and documentation [15]. Data analysis took place after the completion of the second workshop. Combining workshop minutes, workshop notes, observation notes and comments provided in a written format by participants allowed S.S. to identify, organise and report themes using the steps outlined by Braun and Clarke [16]. Key themes were described conceptually and were iteratively discussed between S.S. and T.P.v.S. to ensure no disparities on emerging themes and differences. NVivo 11 (QSR International Pty Ltd., 2014) (QSR International, Doncaster, Melbourne, VIC, Australia) was used to aid in data management and analysis.

## 3. Results

The final response rate for the workshops was 67%. Overall, twenty participants took part in the workshops: ten in Manchester and ten in Nottingham. Participants included a clinical director (n = 1), GPs (n = 13), Practice Managers (n = 2), Prescribing Advisors (n = 3) and a practice member of staff (n = 1). Participants came from different GP practices.

Four key themes emerged through the data analysis: (1) Our day-to-day reality, (2) GPs are competitive, (3) Face-to-face support, and (4) Empowerment and engagement.

### 3.1. Theme 1: Our Day-to-Day Reality

Three examples of the feedback letters were provided for participants to view (see Figure 2, Figure 3 and Figure 4). In discussing these letters, opinions varied about the usefulness and effectiveness of receiving letters about antibiotic prescribing from the CMO. Reflecting on their own experience, most participants had either received or had seen a letter from the CMO. Some participants said that their practice managers had received these letters, who had then cascaded this to all prescribers within the practice, whereas others commented that the GPs had received the letters directly.

All participants agreed that the headline sentence at the start of each letter, usually in a red coloured font, had an impact on gaining the reader’s attention. However, responses to receiving the feedback letters were mixed. On viewing the sample of letters during the workshop, some participants recalled that they had ignored the letter they had received after an initial read due because the data did not reflect the most up-to-date prescribing data, whilst some used the letter as a tool for tackling antibiotic prescribing behaviours within their general practice. Some participants said that they would receive the letter and think about doing something about it; however, day-to-day duties simply got in the way and it was only a matter of days before the letter was forgotten about or binned.

Many participants felt that these letters were important to receive, however, they commented that the data used to inform the letters were often out of date by the time letters arrived at the general practice. In addition, participants who had received letters themselves said that the data used were often from the winter period when prescriptions would naturally increase due to flu and other seasonal factors.

In discussing the ‘next steps’ that were outlined down the right-hand side of the CMO letters, most participants commented that these were not helpful or useful. They said that these ‘next steps’ were already being actioned in many general practices but that they simply did not work or were not in keeping with the reality of day-to-day workloads and overstretched services.

### 3.2. Theme 2: GPs Are Competitive

All participants spoke of the importance of having a personalised approach for each general practice and each GP. Participants discussed that it was not enough for the CMO or the Clinical Commissioning Group (CCG) to send out an automated letter and expect action to be taken, especially when general practices were under pressure to deliver on other targets that were deemed to be more important.

In discussing the use of data in the CMO letters, participants were given three examples to look at. In discussing the letter which presented a graph, participants said that the graphs themselves were not useful. However, if data were going to be used, then these should be the latest available, and not coincide with seasonal increases. Furthermore, letters should be sent out immediately once the latest prescribing data had been analysed. However, several participants reflected on the potential challenges on relying on data when prescribing data itself are subject to inconsistencies and in some cases, manipulation in the way data are coded.

In providing further suggestions on presenting data within the letter, all participants agreed that a localised ‘league table’ would be better than the current graph that is used. The league table could potentially ‘benchmark’ general practices within a localised area, with some participants discussing whether ranking each GP within a practice as a separate table could be beneficial. Participants felt that this needed to be individualised per practice to enable practises to see who is prescribing what, e.g., broad spectrum.

All participants agreed that GPs are competitive and therefore, a GP practice league table would be more effective in changing prescribing behaviours. However, they said that motivations for improving prescribing are focused on mainly being ‘average’ rather than ‘top of the table’ due to the way the financial system works in relation to prescriptions. Therefore, the system itself only motivates GPs to move towards the middle of a league table rather than rise to the top. Some participants suggested that a league table with indicators on where to improve next or with targeted areas for improvement at a practice level would be better than the current three points on what to do next that are featured on some letters.

Participants using the BRIT dashboard were able to provide further insight into how the data in the letter could be better utilised. Rather than the ‘next steps’ on the right-hand side, some BRIT dashboard users suggested that these could be replaced with a breakdown of analysis tailored to the general practice receiving the letter. In addition, BRIT dashboard users discussed the possibility to use analytics to provide one or two core recommendations that would be manageable in its implementation. This could be worded as “This month, we recommend that you look at improving…”.

### 3.3. Theme 3: Face-to-Face Support

All participants felt that a much wider range of support was needed to change prescribing behaviours. This support was not just for the GPs and other prescribers themselves, but support for the practice managers, as well as how the local populations could be engaged with to reduce antibiotic prescriptions requests. Interestingly, one group of participants said that it was the older generation of GPs that they found the hardest to engage with in changing antibiotic prescribing behaviours and not the younger generation of GPs.

Participants who were prescribers commented that they were sometimes having to work with other GPs who did not want to change prescribing behaviours and said that they needed support from outside of the organisations to enable them to do this effectively. Where GPs had been able to engage well with changing prescribing behaviours, it they said that these general practices had a wider range of resources and support to draw up. This included information in a 2-page handout format, education materials on antibiotic prescribing and podcasts.

In addition, there was a mixed response about how much support had been given by CCGs for tackling the issues highlighted in the letter, with some CCGs not providing any support and other CCGs contacting general practices several months after the letters had been received. Some participants stated that they would prefer a face-to-face visit from the CCG, whilst others would prefer to use social media or WhatsApp groups. All participants agreed that a visit from the CCG would provide more influence on changing prescribing behaviours through a focused discussion more than a letter.

### 3.4. Theme 4: Empowerment and Engagement

All participants agreed that Public Health England (PHE) needed to utilise a wider range of engagement options if PHE was serious about reducing prescribing rates for antibiotics. Participants commented that they were under a lot of pressure to meet other targets, such as Quality Improvement and Quality Assurance targets, as well as working with patients in a time- and resource-scarce setting.

Participants said that the way in which the CMO engages with GPs needed to change. Some participants described the current process as feeling ‘out of touch’ with the realities of frontline practice. Prescribers discussed the shift needed by the CMO from ‘telling off’ to the ‘empowerment and education’ of GPs. Suggestions made by participants for engaging with GPs included providing a condensed version of the letter that could be sent to all GPs in a practice through WhatsApp. This should be tailored to each individual general practice to enable the facilitation of discussions within the practice, particularly between a practice manager and a GP. In addition, participants discussed the potential of utilising an antibiotic champion within each practice as someone who could be the key contact for the CCGs. Other participants suggested that a prescribing advisor could be utilised rather than a pharmacist as another key contact person for future engagement with general practices.

Participants discussed the need for regular checks on progression, either through Public Health England or the CCG in a face-to-face capacity. All participants agreed that a more proactive and supportive approach from the CCGs and CMO was needed to really make a difference in reducing prescribing rates. In addition, facilitated discussions with prescribers and CCGs could be used in an ‘educational’ type meeting, where GPs could discuss the issues with their populations, as well as difficulties in prescribing behaviours in a safe environment without judgement.

## 4. Discussion

To our knowledge, this is the first qualitative study to report on how GPs in England have received feedback letters from the CMO on prescribing practices and how they think it could be improved. Four key themes emerged through our data analysis: (1) Our day-to-day reality, (2) GPs are competitive, (3) Face-to-face support, and (4) Empowerment and engagement. In addition, our study was been able to offer further understanding of the types of support that GPs need to facilitate positive behaviour change and the types of prescribing data that could be used in future behaviour interventions.

In discussing the ways in which the letters could be improved, participants agreed that letters should be tailored to each practice and local context. Participants welcomed the use of general practice league table with tailored recommendations that could be implemented easily as an alternative to prescribing data. Providing localised data comparisons would bring the feedback letter to its original iteration when in the first year, this was a comparison to other prescribers in local area teams. However, these were later disbanded and in subsequent years GPs were given feedback compared to the national data.

There are now plenty of data showing that show peer-to-peer comparisons can decrease antibiotic prescribing. For example, the original social norms letter from the CMO of England to GPs reduced prescribing by 3.3%, with the treatment group showing a consistent reduction in prescribing every month for 6-months after receiving the letter, at which point the letter was sent to the control group as well (Hallsworth et al., 2016). A quasi-experimental evaluation of the CMO feedback letter with national comparisons in 2016–17 found a 3.69% decrease in prescribing, so it continued to be effective [10].

Elsewhere, an Australian study involving 6649 GPs tested the impact of letters by the Australian Government’s CMO to high-prescribing GPs [17]. The RCT found that letters containing peer comparison information outperformed the education-only letter and resulted in a 14.8% reduction in antibiotic prescription rates [17]. This suggests personalised approaches to social norm interventions benefit from peer to peer comparisons. Interestingly, the most effective letter used a bar chart. This contradicts the results from our workshops where GPs did not like the bar chart presented in one of the letters that they were presented with. Further social norm interventions in the U.K should test feedback letters with individual GP prescribing behaviour compared to other local GPs versus national comparisons at scale to determine its effect on prescribing behaviour.

Study participants reflected that prescribing decision-making itself is complex and can be determined by a range of factors such as geographical location and population demographics. Future interventions that use social norm feedback should consider tailoring letters to include localised contextual data and providing feedback on individual GP antibiotic prescribing rather than at a practice level. In addition, whilst the original letters were developed using a comprehensive review and analysis of the behaviours that support antibiotic stewardship, and which of the drivers of these behaviours are amenable to change [8], to improve their effectiveness, further research should consider an updated large-scale process evaluation of the letter itself.

In considering the use of a social norms feedback letter to alter GP prescribing behaviour, our study suggests that feedback alone would not have its intended impact on prescribing behaviours when GPs must meet other Quality Improvement and Quality Assurance targets, as well as working with patients in a time and resource scarce setting. Therefore, within primary care, any use of social norms should form part of a suite of approaches aimed at reducing antibiotic prescribing, such as better diagnostic coding, more precise prescribing guidelines and a deeper understanding of appropriate long-term uses of antibiotics [4].

Our study participants agreed that proactive support from PHE and CCGs in the form of general practice visits for discussions and sharing best practice would contribute towards a shift in reducing antibiotic prescribing behaviours. This suggests that there is either a lack of understanding around the roles and responsibilities of the CMO, PHE and the CCGS, or there is a need for the CMO, PHE and CCGs to be more ‘visible’ at lower levels in the health system, such as regionally or locally. Further research is needed to explore this issue.

Similarly, utilising antimicrobial stewardship interventions (for example, an antibiotic champion), as well as engaging with GPs through a range of media such as WhatsApp and Twitter, can facilitate more prudent prescribing behaviours. Future social norm interventions should consider testing the effectiveness of feedback of prescribing using different communication methods.

### Limitations of Study

Our results are based on twenty participants attending one of two workshops in England. The minimum number of participants attending the workshops was a result of GPs dropping out on the day of each workshop due to a combination of seasonal flu and start of the COVID-19 pandemic. This resulted in many GPs and other prescribers not being able to attend the workshops. Therefore, there may be other issues that need to be considered in relation to using feedback letters as a method of changing antibiotic prescribing behaviours.

In addition, due to COVID-19 lockdown, only two workshops in two geographical locations within England were completed. Therefore, our findings may not be generalisable to the rest of England, or the U.K. In applying our findings to primary care, our study has provided further suggestions in how to engage with GPs and other prescribers working within primary care to reduce antibiotic prescribing, such as more proactive support from local CCGs, tailored actions and recommendations for individual GP practices and GPs, and the use of localised and up to date prescribing data in raising concerns in prescribing behaviours.

## 5. Conclusions

In conclusion, the contents of future reminder letters from the CMO should be tailored to each GP and reflect the local context. Consideration should be given to the types of data presented in these letters, with an emphasis on comparing GPs to other local GPs in terms of antibiotic prescribing. Similarly, further work is needed to clarify how, where and when prescribing data are collected with those working in primary care.

Future interventions should consider using a wider range of engagement strategies, rather than the reliance on a single feedback letter approach to reduce antibiotic prescribing within primary care. This should include proactive support from CCGs in the form of general practice visits for discussions, communicating with GPs using digital platforms and social media, and the provision of fora for GPs to share best practice.

There is a need to be clear about the role, function and responsibilities of the CMO, PHE and the CCGs, as well as the different roles and functions of organisations within the NHS and how the wider health system operates, communicated with GPs and others that work in primary care. Future research should explore GP perceptions of the role of the CMO, PHE and CCGs and how these may affect antibiotic prescribing.

## Figures and Tables

**Figure 1 ijerph-18-02602-f001:**
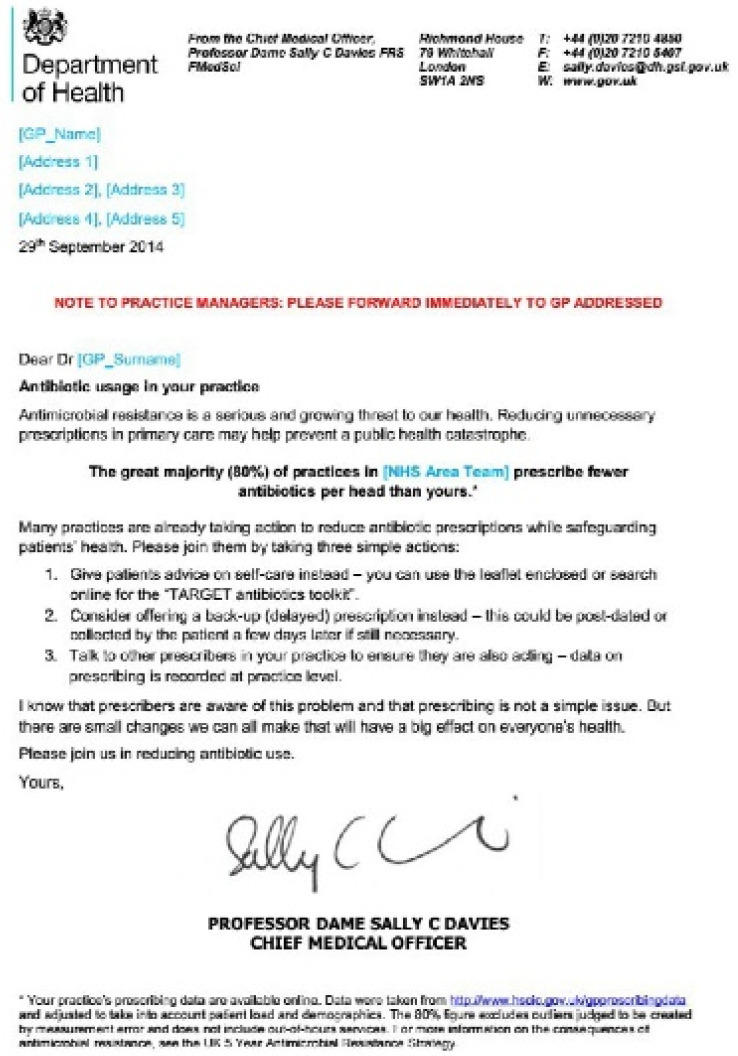
Example of the first GP feedback letter sent to GP Practices in 2014.

**Figure 2 ijerph-18-02602-f002:**
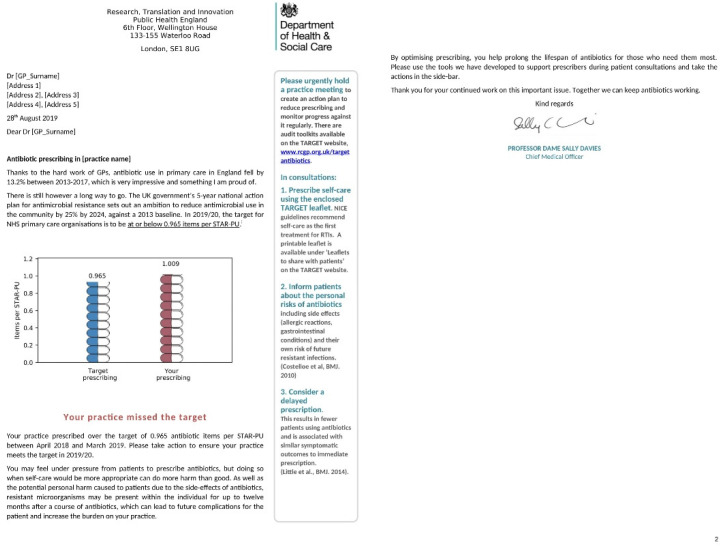
Example 1 of the workshop GP feedback letter with a bar graph in the centre and three key action points boxed to the right of the main text.

**Figure 3 ijerph-18-02602-f003:**
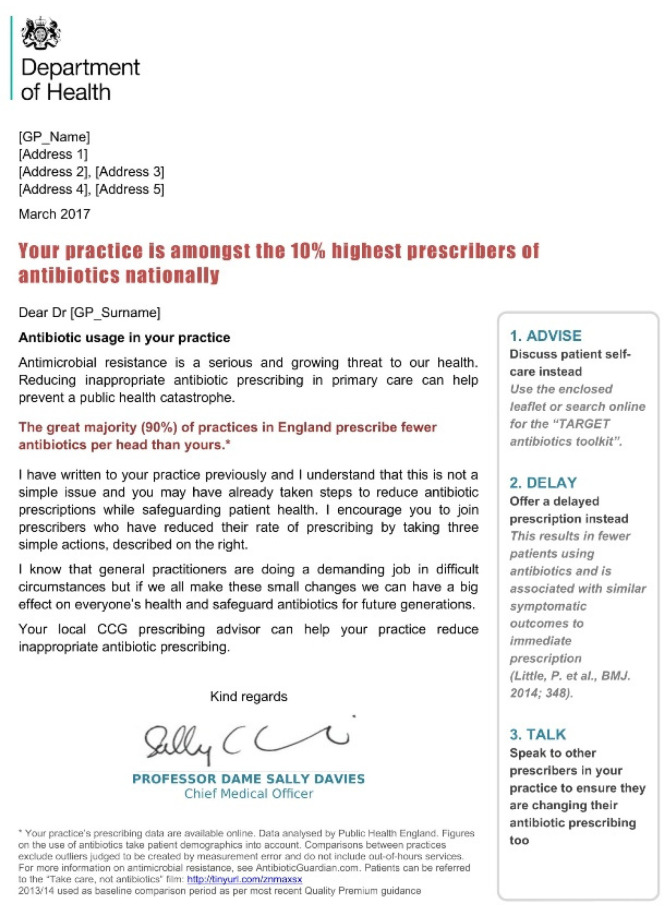
Example 2 of the workshop GP feedback letter with headline result in red at the top of the letter and three key action points boxed to the right of the main text.

**Figure 4 ijerph-18-02602-f004:**
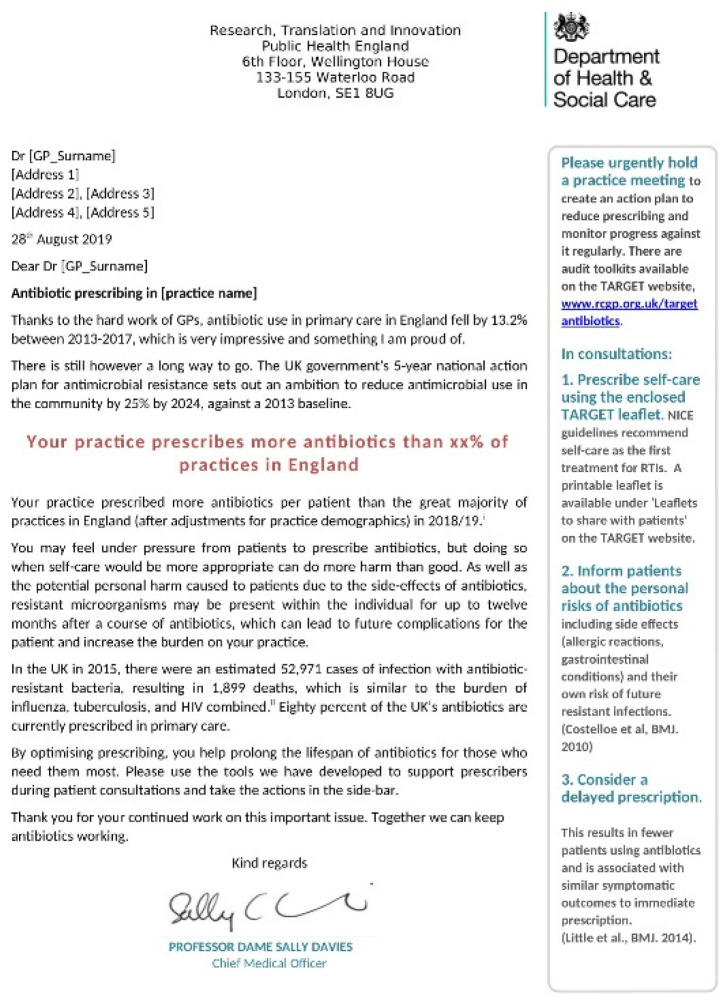
Example 3 of the workshop GP feedback letter with headline result in red in the middle of the letter and three key action points boxed to the right of the main text.

## Data Availability

Data available on request due to restrictions.

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
