# Peer review of "Improving Our Understanding and Practice of Antibiotic Prescribing: A Study on the Use of Social Norms Feedback Letters in Primary Care"

_ijerph, 2021, doi:10.3390/ijerph18052602_

Round 1

Reviewer 1 Report

This is an extremely interesting paper, adopting a qualitative participatory design to explore alternative approaches to providing feedback to GPs to reduce antibiotic prescribing. Recommendations are in bold below.

The introduction provides a clear rationale for the study. Some definition of ‘social norm intervention’ would be helpful.

The sample is perhaps limited by drawing participants from only two nearby geographical areas (Manchester & Nottingham). It would be helpful to clarify whether the 10 participants in each of the two workshops all came from different GP practices, or whether they may reflect involvement of a smaller number of practices. Some description of the characteristics of the participating GP practices would be helpful e.g. the level of antibiotic prescribing; demographics of GPS (if available), populations served by the included practices in relation to characteristics which are known to influence patient demand or GP behaviour i.e. discuss how typical / atypical participating GP practices may be.   

Whilst the methods used are described as thematic analysis, the resultant themes appear to simply reflect the research aims / topic guide, without the level of interpretation and conceptualisation normally associated with Braun & Clark’s method; it would appear that more of a content analysis approach was adopted. The authors should perhaps consider modifying their description of the qualitative method used.

Overall, the presentation of results is clear and informative, with insightful and constructive suggestions offered by participants. However, as noted above, the resultant ‘themes’ seem to reflect the topic guide, rather than a conceptualisation of key issues e.g. phrases used such as ‘our day-to-day reality’, ‘GP’s are competitive’ ‘face-to-face support’, and ‘empowerment & engagement’ may provide more conceptual ‘punch’ to describe themes, giving the reader a more immediate understanding of the key findings.

The discussion relates key findings to existing literature & makes appropriate recommendations for future research. It might have been helpful to discuss how behavioural science could be applied to the findings & recommendations; the abstract alludes to the use of behavioural science, however, this was largely absent from the discussion.

References 7 & 9 appear to refer to the same output?

Typos:

Line 54 – that/than

line 76 – times/types

line 107 – the figure is reversed

line 296 – ‘found a 3.69% it caused decrease’

Author Response

Dear Reviewer 1, 

Thank you for the constructive comments regarding our manuscript “Improving our understanding and practice of antibiotic pre-scribing: a study on the use of social norms feedback letters in primary care”.

We have revised our manuscript according to your suggestions. Changes to the manuscript have been completed in track changes using the manuscript provided by IJERPH. 

Reviewer 1 suggested that a definition of ‘social norm intervention’ would be helpful in the introduction. We have included a definition for further clarity.

Reviewer 1 suggested that further information was added with regards to the participants who added the workshops. We have added a sentence to explain that participants came from different GP practices. However, we are not able to provide a breakdown of individual GP antibiotic prescribing, patient demand or GP behaviour, for all participants. Therefore, it was felt that as we cannot provide all this information for all GPs taking part, we should not include it.

Reviewer 1 commented that as the data anaylsis was not conducted in as much depth as outlined by Braun and Clark, we should consider changing our data analysis to a content analysis. Furthermore, it was noted that the presentation of the results and resulting themes appear to reflect part of the research aims. It was not intentional that the themes appear to reflect part of the research aims. As noted in our limitations section, our level of interpretation and conceptualisation was limited by the number of participants and method of data collection. As we used Braun and Clark’s method, it would be difficult to state that we used another method for analysis when we did not. We have therefore considered the recommendations for changing the themes and have agreed to use the reviewer’s recommendations.

Reviewer 1 noted that it would be helpful to discuss how behavioural science could be applied to the findings and recommendations, as this was noted in the abstract. We have altered the abstract to reflect that the focus of the study is on social norms. We have made it more explicit in the discussion section that we are focussing on social norms.

Reviewer 1 queried if references 7 and 9 refer to the same output. Reference 7 is a report of the output and reference 9 is the academic paper that was published. There are elements missing from the report that only appear in the academic paper, hence the use of different references.

Reviewers 1 noted several typing errors in the manuscript. These have been corrected.

Reviewers 1 noted that Figure 3 is mirrored. This has been amended.

Thank you once again for your constructive comments.

Reviewer 2 Report

This is an interesting article and I agree with the authors in that incorporation of the behavioral sciences is increasing more important in antimicrobial stewardship initiatives.  It is unfortunate that recruitment was affected like it was due to COVID.

I have only minor recommendations after reviewing this manuscript.

In the introduction, the last sentence in the 1st paragraph, "Therefore, there is a need to better understand how different prescribing practices can impact AMR" seems a little out of place for this paper. This is really the only mention of AMR and your purpose of this study is not evaluating resistance but essentially gathering feedback on providing feedback to prescribers.

Figure 3 is mirrored

Author Response

Dear Reviewer 2, 

Thank you for the constructive comments regarding our manuscript “Improving our understanding and practice of antibiotic pre-scribing: a study on the use of social norms feedback letters in primary care”.

We have revised our manuscript according to your suggestions. Changes to the manuscript have been completed in track changes using the manuscript provided by IJERPH. 

Reviewer 2 noted several typing errors in the manuscript. These have been corrected.

Reviewer 2 suggested the following sentence "Therefore, there is a need to better understand how different prescribing practices can impact AMR" was out of place. We have removed the sentence.

Reviewer 2 noted that Figure 3 is mirrored. This has been amended. 

Thank you once again for your constructive comments.